# Peer J

# Comparing two models that reduce the number of nephrology fellowship positions in the United States

Tejas Desai

Division of Nephrology and Hypertension, East Carolina University, Greenville, NC, USA

## ABSTRACT

There has been a steady decline in the number of applications to nephrology training programs. One solution is to decrease the number of available fellowship positions. Proponents believe that training programs have grown too big but the method for reduction has not been established. This investigation analyzes two models that decrease the number of available training positions and compares them head-to-head to identify the least burdensome method by which this reduction should occur. In the survival of the fittest model (SotFM) fellowship positions are eliminated if they were unfilled in the National Residency Match Program's (NRMP) 2013 Specialty Match. In the equal proportions model (EPM) a formula is used to calculate a priority score using ESRD prevalence data from the 2013 USRDS Report and the geometric mean between a given jurisdiction's current apportionment ($n$) and its next position ($n + 1$). The least burdensome model is that which results in the (1) least number of jurisdictions losing fellow positions and (2) lowest percent reduction for any single jurisdiction. There were 416 nephrology positions offered and 47 unfilled in 2013. In the SotFM, 23 jurisdictions would sacrifice these 47 positions. In the EPM, 369 positions were apportioned (=416–47); only 9 jurisdictions would experience a reduction. The largest single-jurisdiction reduction in fellow positions was 67% (SotFM) and 50% (EPM). The EPM results in a less burdensome reduction of fellow positions nationwide. The EPM is a time-tested model that injects fairness into the painful process of reducing the total number of fellow positions across America.

## INTRODUCTION

There has been much debate regarding the decrease in applications to nephrology training programs. For more than a decade, the number of applications has steadily dropped while the number of explanations and solutions for this decline has risen (*Desai et al., 2012*; *Jhaveri et al., 2013*). These solutions are as varied and unique as the individuals who propose them. One commonly proposed solution is to simply decrease the available number of fellow positions (*Desai, 2014*). Proponents argue that fellowship programs have grown too big and have overestimated the interest that resident physicians have in nephrology careers (*Berns et al., 2014*). From 2009 to 2013, the applicant-to-position ratio, a common metric used to characterize the demand for positions, has steadily fallen from

Corresponding author
Tejas Desai,
Tejas.p.desai@gmail.com

1.6 to 1.01 (2010: 1.5; 2011: 1.3; 2012: 1.1) (*National Resident Matching Program, 2013*). Many influential voices, including leaders of the American Society of Nephrology (the largest professional society of Nephrologists) have voiced this recommendation through formal reports (*Berns et al., 2014*; *Salsberg et al., 2014*). However, there is no consensus on how the excess positions should be eliminated. In this investigation, we analyze two models that decrease the number of available training positions and compare them head-to-head to identify the least burdensome method by which this reduction should occur.

## METHODS

### Data set

The two models rely on two separate data sets. The first model, known as the Survival of the Fittest (SotFM), uses data from the *National Resident Matching Program (2013)*'s (NRMP) Specialty Match Report for Nephrology. This report contains statewide information about available fellowship positions, the number of applicants, and the number of unmatched positions. The second model, known as Equal Proportions (EPM), uses data from the same NRMP Specialty Match Report along with 2011 statewide prevalence of end-state renal disease (ESRD) patients from the 2013 United States Renal Data System (USRDS) Report (*U.S. Renal Data System, 2013*).

### Survival of the fittest model (SotFM)

In this model, the number of fellowship positions that should be eliminated is equal to the number of unmatched positions in the 2013 NRMP Specialty Match. Briefly, since 2009 the NRMP has used its proprietary and confidential matching algorithm to pair an applicant with his/her single most favored (highest ranked) training program (http://www.nrmp.org/). Positions that were not allocated to any applicant through the use of this matching algorithm are labeled unmatched (unfilled, vacant). We calculated the number of unmatched positions by jurisdiction (states and the District of Columbia) after the completion of the 2013 Specialty Match (4 December 2013) and deducted that number from the total number of fellowship positions nationwide. This SotFM model is commonly referred to as "attrition" or "natural selection".

### Equal proportions model (EPM)

In this model, the total number of fellowship positions per jurisdiction is mathematically calculated from the number of prevalent ESRD patients for each state respectively. Historically, a variant of the current version of the EPM has been used to apportion the total number of a state's legislators to the United States House of Representatives since 1789 (i.e., Hamilton-Vinton, Adams, Dean, Webster, Jefferson and Hill methods) (*Schmeckebier, 1952*; *Crocker, 2010*; *Congressional Apportionment*). The EPM calculates the total number of legislators for each state from that state's total population. The model was codified in 1929 (2 U.S.C. 2a) and the United States Congress has used the current formula since 1941 (*The Permanent Apportionment Act of 1929*). Briefly, the EPM formula begins by apportioning one position to each jurisdiction (*Congressional Apportionment*). Subsequent positions are apportioned based on a calculated priority score. The score is derived by using the

prevalence of ESRD patients in each jurisdiction and the geometric mean between a given jurisdiction's current number of fellowship positions ($n$) and its next apportioned position ($n + 1$). This iteration continues until the total number of fellowship positions nationwide has been apportioned.

In this investigation, two important rules of apportionment were implemented. First, any jurisdiction whose total number of fellowship positions was zero in the 2013 NRMP Specialty Match (quota) was excluded from the model. Fellowship positions were not apportioned to any of these jurisdictions. Second, no jurisdiction could be apportioned a total number of fellowship positions greater than its 2013 NRMP quota.

### Identifying the least burdensome model

Both models were compared head-to-head in two areas. First, the number of jurisdictions that would experience a decrease in their total number of fellowship positions (based on the 2013 NRMP quota for each jurisdiction) was determined. Second, the greatest percent reduction that a jurisdiction would experience was calculated. The model that resulted in (1) the least number of jurisdictions losing fellow positions and (2) the lowest percent reduction for any single jurisdiction was deemed "least burdensome." The models were considered equally burdensome if they each shared one of these two features.

### Statistical considerations

Microsoft Excel 2011 (Microsoft, Redmond, Washington) was used to calculate priority scores (see EPM above). JMP Pro 10.0 (SAS, Cary, North Carolina) was used to calculate median, interquartile ranges and generate figures and tables.

### RESULTS

In 2013 there were 8 jurisdictions that had zero positions in the NRMP Specialty Match: Alaska, Hawaii, Idaho, Montana, Nevada, North and South Dakota, and Wyoming. These jurisdictions were excluded from this investigation. The remaining 43 jurisdictions (42 states and the District of Columbia) offered a total of 416 positions (Fig. 1). There were a total of 421 applicants for an applicant:position ratio (AP ratio) of 1.01. Forty-seven positions were unmatched and resulted in an 89% match rate.

In the SotFM, these 47 unmatched positions were eliminated from 23 jurisdictions (53% of the total jurisdictions participating in the NRMP Specialty Match). Figure 2 shows the distribution of jurisdictions losing $n$ percent of their quota. The median percent reduction was −20.00% (IQR −12.50%, −37.50%). In the EPM, a total of 369 positions were apportioned to 43 jurisdictions. Each jurisdiction was apportioned 1 position at the start and none were apportioned a total number of positions greater than their 2013 NRMP quota. Table S1 shows the priority scores for each jurisdiction during each of the 368 rounds of apportionment. Nine jurisdictions were apportioned less fellow positions than their NRMP quota (21% of the total jurisdictions participating in the NRMP Specialty Match). Figure 3 shows the distribution of jurisdictions losing $n$ percent of their quota. The median percent reduction was −33.33% (IQR −24.09%, −50.00%).

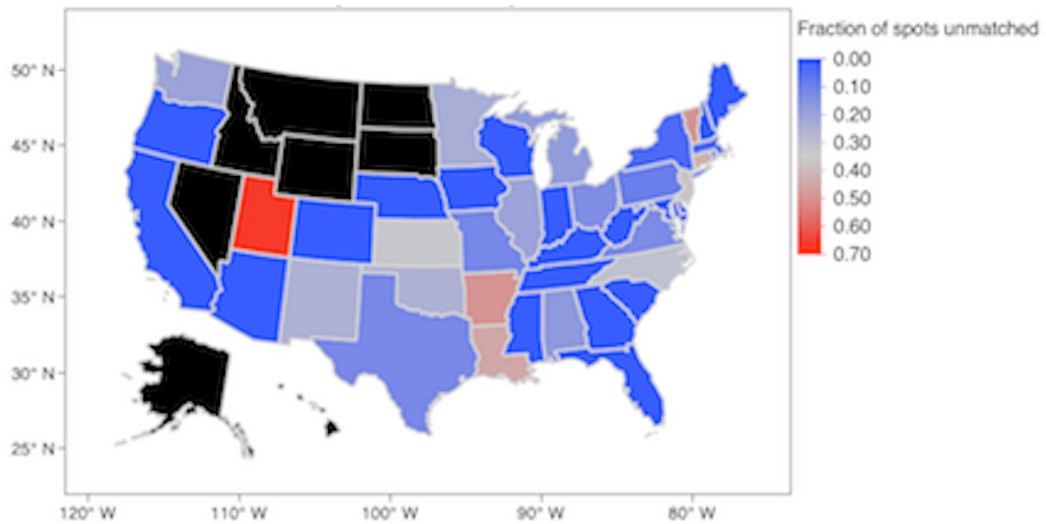

**Figure 1** Fraction of unmatched spots in the 2013 NRMP Specialty Match for Nephrology.

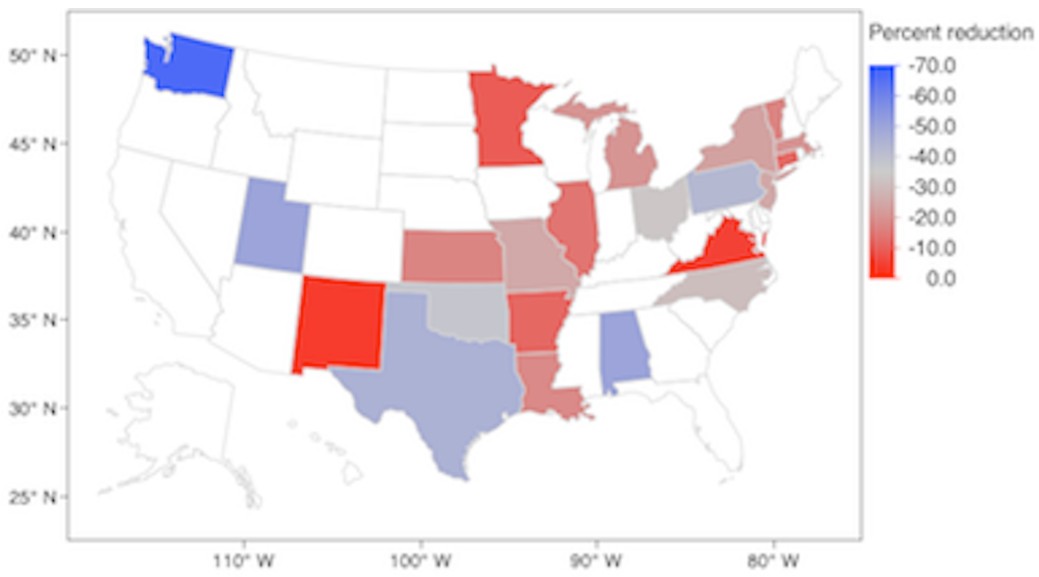

**Figure 2** Jurisdictions experiencing a reduction in fellowship positions (percent) in the Survival of the Fittest Model. Uncolored states experienced a zero percent reduction or had a zero quota in the 2013 NRMP Specialty Match.

In the SotFM, Washington suffered the greatest reduction in positions (−66.67%) followed by Utah and Alabama (−50.00% each). In the EPM, Alabama, Vermont, and Virginia suffered the greatest reductions (−50.00% each).

## DISCUSSION

Decreasing the number of nephrology fellowship positions will be a challenging, arduous, and difficult process regardless of the method implemented to carry out that reduction. The field of nephrology (as a whole) must identify a model that minimizes the systemic

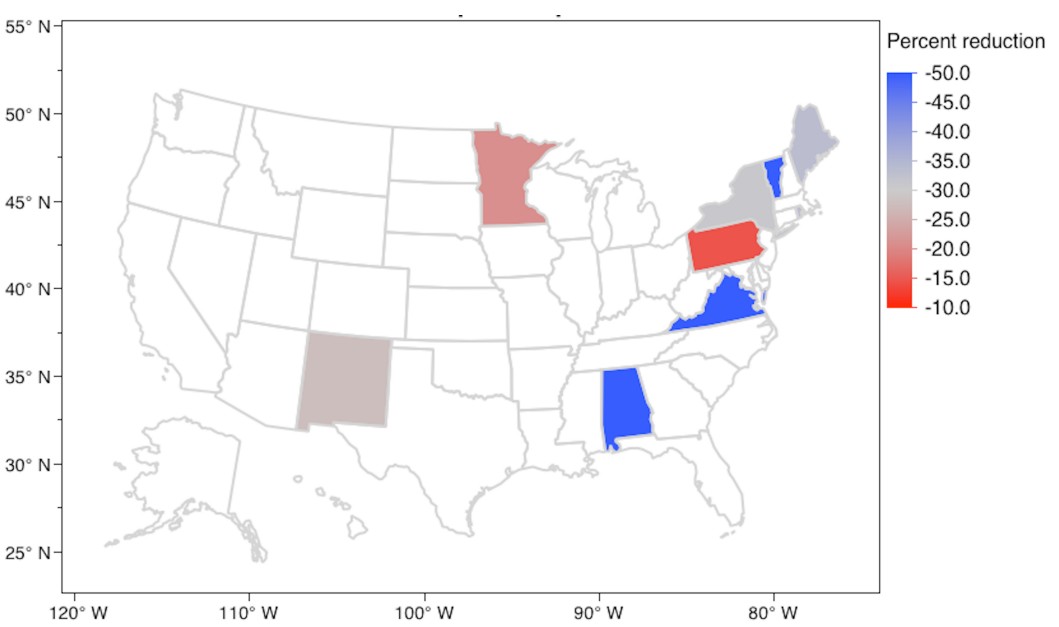

**Figure 3 Jurisdictions experiencing a reduction in fellowship positions (percent) in the Equal Proportions Model.** Uncolored states experienced a zero percent reduction or had a zero quota in the 2013 NRMP Specialty Match.

burden that will occur when reduction takes place. In this investigation, the EPM reduces the systemic burden in four ways. First, it limits the reductions to only 9 jurisdictions (less than a quarter of all jurisdictions) versus 23 jurisdictions (over half of all jurisdictions) seen in the SotFM. Second, the greatest single reduction in the EPM is only 50%, while it is 67% in the SotFM. Third, the method by which the EPM was implemented in this investigation ensured that no jurisdiction would benefit at the expense of another. Jurisdictions that did not offer fellowship positions in 2013 were not "awarded" fellowship positions in the EPM. Moreover, those jurisdictions that offered positions in the 2013 NRMP Specialty Match could not increase their total number of positions at the expense of another jurisdiction. Finally, the EPM reduction model eliminates positions based on patient needs (i.e., the prevalence of ESRD patients) and not on the perception of a fellowship program's prestige and value.

An additional downfall of the SotFM is that it eliminates fellowship positions based on the assumption that applicants avoid positions from less prestigious programs. While the SotFM doesn't require the use of a mathematical formula and does not account for patient needs, its execution is arbitrary and should not be supported. Applicants may not choose certain fellowship positions because of reasons that don't reflect the quality of the fellowship program, such as climate patterns, geopolitical concerns, employment opportunities for spouses, or proximity to family. The SotFM cannot disconnect the personal preferences of an applicant from the educational value or prestige of the program that offers the position.

Lastly, reducing fellowship positions will not directly increase interest in the field of nephrology. At a minimum, reducing the number of positions offered nationwide

Table 1 NRMP Specialty Match Data for 2-year fellowship training programs.

| Fellowship | Applicants | Positions | AP* Ratio | Matched positions (%) |
|---|---|---|---|---|
| Nephrology | 421 | 416 | 1.01 | 89 |
| Endocrinology | 361 | 251 | 1.44 | 95 |
| Rheumatology | 244 | 195 | 1.25 | 95 |
| Infectious disease | 288 | 334 | 0.86 | 81 |
| Pulmonary | 96 | 22 | 4.4 | 100 |
| Oncology | 120 | 31 | 3.9 | 94 |
| Hematology | 99 | 15 | 6.6 | 100 |

Notes.
* Denotes applicant:position ratio.

will make it easier for fellowship programs to match applicants. Historically, fellowship programs with applicant:position ratios (AP) greater than 1.0 have had greater success in filling all positions (*Subspecialty Match Report, 2014*). Successful 2-year fellowship training programs such as Rheumatology or Endocrinology have AP ratios of 1.25 and 1.44, respectively, while another struggling 2-year training program, Infectious Disease, has an AP ratio of 0.89 (*National Resident Matching Program, 2013*). Eliminating 47 nephrology fellowship positions would bring the AP ratio from 1.01 to 1.14 (Table 1). A more successful match can have positive consequences; it would allow training programs to devote resources away from the post-match "scramble" and towards more educationally valuable endeavors (*Desai, 2014*).

## Additional considerations

At the time of this writing, there was no central authority that could mandate a decrease in nephrology positions. Therefore, this investigation serves only as a guide for training programs across the United States. It allows programs to collectively manage the number of available positions and limit the overall burden experienced by a reduction. However, recent recommendations by the Institute of Medicine suggest that a centralized agency, referred to as the "Graduate Medical Education (GME) Center", could develop and enforce a funding plan that controls the number of physicians required to meet the nation's healthcare needs (*Wilensky & Berwick, 2014*). The results derived from the EPM would allow such an agency to resourcefully use GME funding to achieve this goal.

It is unclear if other models of reduction exist that can be compared to the 2 models illustrated in this investigation. Alternative models were not found in the literature search preceding the start of this investigation. Arguably a third model would mathematically link the number of fellowship positions (by jurisdiction) to the number of all patients with kidney disease, not just those receiving dialysis. However, there is no centralized database that monitors the total number of patients cared for by kidney doctors. Such a model would rely on internal institutional reports; reports that could be incomplete, inaccurate, and/or not easily validated.

Neither the SotFM nor EPM accounts for positions that are available "outside" of the NRMP Specialty Match program. There is no central agency that tallies these "non-match"

positions; if there are many non-match positions then both models could underestimate the burden that each jurisdiction would experience.

No published data was found that revealed the practice location preferences of graduating nephrology fellows. Educators in nephrology who believe that many graduating fellows do not practice within the same state in which they trained are relying on personal observations or internal institutional data—neither of which is published or generalizable. The only data available is from Internal Medicine graduates; jurisdictions have reported anywhere from 41% to 60% of graduates remaining within their state to begin independent practice (*Seifer, Vranizan & Grumbach, 1995*; *Owen, Hayden & Bowman, 2005*; *Packham et al., 2012*; *Tucker et al., 2012*; *MMS Physician Workforce Study, 2013*; *Rumack, 2013*). Such a high proportion of graduates remaining in the state in which they trained supports the EPM.

The reader may conclude that positions from "prestigious" programs would be reduced. At face value, one may believe the EPM offers less of a choice for applicants. One would erroneously assume that a model, which reduces positions at "prestigious programs", would be a disservice to the applicant. The EPM, however, does not discriminate between programs considered to be "more prestigious" from those considered to be less. The goal of the EPM is to reduce positions in an objective manner by linking the workforce needs (i.e., fellowship positions) with patient needs (i.e., ESRD prevalence). This method does not factor "prestige" into the mechanism of reduction. Indeed, prestige is subjective: Who can decide which program is more or less prestigious when all programs must deliver healthcare to ESRD patients as prescribed by national guidelines? Prestige is transient; prestigious programs today may become less prestigious tomorrow and vice versa. Applicant demand for positions in a particular program doesn't necessarily reflect the prestige of said program. Applicants consider other factors besides program prestige when submitting applications; those factors include but are not limited to (1) political issues (e.g., states that are for/against same-sex marriage), (2) weather, (3) proximity to family/friends, and (4) employment opportunities for spouses. The fact that the EPM is blind to all subjective factors, such as prestige, state politics, weather, etc., supports the conclusion that it is an equitable model.

The EPM model described in this investigation is operationalized at the state level. The calculations are not granular enough to dictate how or which specific training programs should reduce their number of positions. As in the apportionment of representatives to the US House, decisions regarding the distribution of positions across programs of a specific jurisdiction should remain a "local" one. The EPM model allows program directors within a jurisdiction to reach a consensus in how and where positions should be reduced. For example, program directors can collaborate to reduce positions equitably and/or preferentially from particular tracks (research over clinical). In the EPM, jurisdictions are afforded the flexibility they need to successfully reduce the number of training positions.

## CONCLUSION

In this investigation, two models were compared to determine which provided the least burden to the nation when reducing the number of Nephrology fellowship positions. The Equal Proportions Method (EPM), historically used to apportion legislators to the US House since 1789, mathematically links the needs of the ESRD population to the number of fellowship positions offered. The EPM is blind to the prestige of a fellowship training program or to the personal (non-professional) preferences of an applicant. This model should be considered and implemented if the field of Nephrology wishes to improve its matching rate.

### Funding

The author declares there was no funding for this work.

### Competing Interests

Tejas Desai is the creator of Nephrology On-Demand Plus (http://goo.gl/tfSAQT) and the Nephrology Fellowship Director at East Carolina University. The data reported here have been supplied by the United States Renal Data System (USRDS). The interpretation and reporting of these data are the responsibility of the author and in no way should be seen as an official policy or interpretation of the U.S. government.

### Author Contributions

- Tejas Desai conceived and designed the experiments, performed the experiments, analyzed the data, contributed reagents/materials/analysis tools, wrote the paper, prepared figures and/or tables, reviewed drafts of the paper.

### Supplemental Information

Supplemental information for this article can be found online at http://dx.doi.org/10.7717/peerj.720#supplemental-information.

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
