# Peer review of "Comparing two models that reduce the number of nephrology fellowship positions in the United States"

_PeerJ, doi:10.7717/peerj.720_

## Round 0.1 · original submission · Minor Revisions

· Academic Editor

Minor Revisions

Please follow the comments of the three reviewers and address particularly the comments of Reviewer #3. Also, citing of tweets should be avoided. NLM style for references (http://www.ncbi.nlm.nih.gov/books/NBK7256/) does not currently include tweets.

·

Basic reporting

The manuscript adheres to PeerJ policies, is written in English (with an occasional grammatical error - eg line 82, ... "had a zero positions"), and is structured according to the Guidelines.
The manuscript cites relevant literature and demonstrates how it fits into the broader field of understanding of the topic, the Figures and Tables are relevant to the content.

Minor comments:
1) The Introduction portion may benefit from an inclusion (and further elaboration on) of a part of the "Additional considerations" (lines 137-138) portion of the manuscript, to make sure the readers understand that this manuscript is an intellectual exercise, as there is no governing body (yet) that can benefit from the presented results.
2) The references do not adhere to the guidelines.
3) Occasional grammatical errors.
4) The names of the manufacturers of software used in the data analysis are not mentioned.

Experimental design

The experimental design is sound, and the investigation has been conducted rigorously. The methods have been described in enough detail (notwithstanding the full identification of used software).

Validity of the findings

The analysis is valid and sound, and the data used for evaluation and analysis are publicly available. The conclusions are logically tied to the research questions, and are supported by the evaluated data.

Additional comments

Minor comments:
1) The Introduction portion may benefit from an inclusion (and further elaboration on) of a part of the "Additional considerations" (lines 137-138) portion of the manuscript, to make sure the readers understand that this manuscript is an intellectual exercise, as there is no governing body (yet) that can benefit from the presented results.
2) The references do not adhere to the guidelines.
3) Occasional grammatical errors.
4) The names of the manufacturers of software used in the data analysis are not mentioned.

·

Basic reporting

I would suggest that Figures should be produced in different shades of white, gray and black (instead of different colours). If the article is printed out in black and white, the colour Figures are unintelligible.
It seems that Priority scores for each jurisdiction was not intended to be a regular part of the article, but as an appendix. In this case, it should not be labeled as Table 1, but as an Appendix or Supplemental material. Please consult the instructions for authors for the appropriate format.
I would recommend the authors to avoid citing Tweets in a scientific article.
"Additional Consideration" is not a standard section of the PeerJ, please consider integrating this section with the DIscussion.

Experimental design

No Comments.

Validity of the findings

I would suggest the authors to provide the NRMP Specialty Match applicant:position ratio for the last 5 years (not only for 2013). This would indicate a temporal trend and would be instructive for readers.
In the Conclusion it is stated that EPM is "blind to the prestige of a fellowship training program or to the personal prefrences of an applicant". This feature of EPM should be considered more carefully and addressed appropriately. It seems that EPM model does not take into account the fact that the applicants will still have their less preferred fellowship training programs and that such programs may eventually remain unfilled, regardless of the model of reducing the number of fellowship positions.

Reviewer 3 ·

Basic reporting

I have no concerns with the basic reporting methods. It is a well written manuscript and is clear throughout.

Experimental design

Reasonable.
I have some comments on the EPM model:
I am not sure than ESRD rate in a particular state is a fair reflection of demand for fellow positions. There is a whole lot more to Nephrology that just provision of ESRD services. In fact, outpatient dialysis likely constitutes a minority of work for renal fellows. Activity of the transplant program, CKD numbers, other nephrology specialty services provided etc. could all be relevant.
The point can be made that dialysis provision will be a large proportion of work for the graduating fellow and this is true. However, many (?most) fellows will not practice in the state that they trained.
Even within states there may be great variability in prevalence of ESRD which so that just because a certain program is located in a state with high/low ESRD prevalence, it doesn't necessarily follow that the catchment area of the hospital will be in an area of high/low ESRD prevalence.

Validity of the findings

The author has reported his findings which demonstrate that the EPM method of reducing fellowship positions appears 'least burdensome'. This is certainly true for overall programs but some programs (?more 'prestigious' programs) will fare worse with this model compared to the SofFM. Moreover, what is good for programs is not necessarily good for trainees. As we need to encourage doctors in to the specialty we should not restrict when they want to train.

The underlying issue is that we need to make Nephrology more attractive (easier said than done I appreciate). Reducing training positions may make matching easier for the programs but will this really solve anything?

Additional comments

Well written, well researched piece from an interested author with a track record in this area. If fellowship numbers are to be reduced then this sort of thinking needs to be embraced. Despite this, I feel the EPM model described in this paper make be criticized as being too simplistic by existing programs directors if/when it comes to reducing positions.

Minor comments:
Line 68 'Second, no jurisdiction could be apportioned a total number of fellowship
positions greater than its 2013 NRMP quota'...please explain why?
Line 120: Double negative here is confusing to the reader.

---

## Round 0.2 · Minor Revisions

· Academic Editor

Minor Revisions

One of the reviewers has provided clarification of the comments in the review, so that you can address them in the final version of the manuscript.

·

Basic reporting

No Comments.

Experimental design

No Comments.

Validity of the findings

No Comments.

Additional comments

The Manuscript has gained on clarity and readability.

·

Basic reporting

No Comments.

Experimental design

No Comments

Validity of the findings

My initial comment:
In the Conclusion it is stated that EPM is "blind to the prestige of a fellowship training program or to the personal prefrences of an applicant". This feature of EPM should be considered more carefully and addressed appropriately. It seems that EPM model does not take into account the fact that the applicants will still have their less preferred fellowship training programs and that such programs may eventually remain unfilled, regardless of the model of reducing the number of fellowship positions.

Author’s answer:
Dear Dr. Sambunjak: Is it possible for you to clarify this point further? I have read this specific recommendation and the Conclusion but I am not able to understand what the error or misunderstanding is. If you could help me out in understanding what you want me to do, I would be happy to make the appropriate changes. Thanks and sorry for not understanding your recommendation.

My clarification:
EPM model, which in the Conclusion section is recommended as preferable to the SoTF model, calculates the total number of fellowship positions per jurisdiction from the number of prevalent ESRD patients for each state respectively. Also, it is stated that EPM model is “blind to the prestige of a fellowship training program or to the personal preferences of an applicant”, which is exactly what may be the problem with this model. I presume that currently there are some fellowship programs in the USA which are prestigious and sought after – these programs (again, I presume) have a higher applicant-to-position ratio and so they have to turn down a considerable number of applicants. On the other hand, some less prestigious programs probably have A/P ratio even lower than the 2013 average of 1.1. This means that they cannot fill all of their available positions with appropriate applicants. If you apply the EPM model, which doesn’t take into account the demand for positions at specific fellowship programs, you may end up with the reduced number of positions in the prestigious programs and an increase of the number of positions at the less prestigious programs. However, one cannot presume that those applicants who were turned down at the more prestigious programs will go and apply for the less prestigious programs (rather, they may apply for the more prestigious programs in another specialty). The result may be that the more prestigious programs (now with the less positions available) will still have a high A/P ratio and will be fully “booked”, whereas the less prestigious programs (some of them now with an increased number of available positions) will have an even lower A/P ratio (due to the increase in P, without an increase in A). Consequently, you will have overbooked (highly prestigious) programs, non-viable (less prestigious) programs, and a decrease in the number of nephrology specialists which will still not correspond with the prevalence of ESRD patients. This should be addressed in the Conclusion section.

Reviewer 3 ·

Basic reporting

No comments

Experimental design

No comments

Validity of the findings

No comments

Additional comments

While the model may not be ideal (and no model may be ideal), and it is not a solution to the problem, it is valid research and worthy of publication. The crisis facing Nephrology is complex with many causes and therefore many actions necessary. This seeks to address one facet of the problem namely supply/demand of training posts. My comments have been addressed in the rebuttal. The author has previously considered the issues and with data available, this is the best model available. It warrants wider discussion in the nephrology community even if this reviewer doesn't fully agree with its implementation in its current form.

---

## Round 0.3 · accepted · Accept

· Academic Editor

Accept

Thank you for the clarification and final changes to the manuscript.